# Compositional Models: Multi-Task Learning and Knowledge Transfer with Modular Networks

## Abstract

Conditional computation and modular networks have been recently proposed for multitask learning and other problems as a way to decompose problem solving into multiple reusable computational blocks. We propose a novel fully-differentiable approach for learning modular networks. In our method, the modules can be invoked *repeatedly* and allow knowledge transfer to novel tasks by adjusting the order of computation. This allows soft weight sharing between tasks with only a small increase in the number of parameters. We show that our method leads to interpretable self-organization of modules in case of multi-task learning, transfer learning and domain adaptation while achieving competitive results on those tasks. From practical perspective, our approach allows to: (a) reuse existing modules for learning new task by adjusting the computation order, (b) use it for unsupervised multi-source domain adaptation to illustrate that adaptation to unseen data can be achieved by only manipulating the order of pretrained modules, (c) show how our approach can be used to increase accuracy of existing architectures for image classification tasks such as IMAGENET, without any parameter increase, by reusing the same block multiple times.

## 1 Introduction

Most modern interpretations of neural networks treat layers as transformations that convert lower-level features, such as pixels, into higher-level abstractions. Adopting this view, many designs that break up monolithic models into interacting modular components localize them to particular processing stages thus disallowing parameter sharing across layers. This restriction, however, can be limiting when we need to continually grow the model, be that for transferring knowledge to a new task or preventing catastrophic forgetting. It also prohibits module reuse within a model even though certain parameter-efficient architectures and complex decision making processes may involve recurrent components (Kaiser & Sutskever, 2016; Randazzo et al., 2020) and feedback loops (Herzog et al., 2020; Yan et al., 2019; Kar et al., 2019).

Here we propose a simple alternative design that represents an entire model as a mixture of modules each of which can contribute to computation at any processing stage. As opposed to many other approaches (Zoph & Le (2016); Bengio (2016); Kirsch et al. (2018a); Rosenbaum et al. (2019) etc.), we use a soft mixture of modules to obtain a more flexible model and produce a differentiable optimization objective that can be optimized end-to-end and does not involve high-variance estimators. Following our approach, the parameters of every block (or layer) of the network is computed as a linear combination of a set of "template" block parameters thus representing the entire model as: (a) a databank of template blocks and (b) vectors of "mixture weights" that are used to generate weights for every layer. This simple design can be utilized for a variety of applications from producing compact networks and training multi-task models capable of re-using individual modules to knowledge transfer and domain adaptation. The experimental results show that: (a) when used for multi-task training, our model organizes its modules where tasks share first few layers while specializing closer to the head, while (b) in domain adaptation problems modules instead specialize on processing the image, while sharing later processing stages. Moreover, our self-organizing model achieves promising results in multi-task learning and model personalization. The rest of the paper is organized as follows: in Section 2 we go over the related literature and discuss the existing approaches to modular

networks and neural architecture search; in Section 3 we describe the architecture of our conversational model and various aspects of learning the best module sequence for each specific task; we introduce the experimental results in a single-task setting, multi-task learning, continual learning and domain adaptation in Section 4 and provide a final discussion in Section 5.

## 2 PRIOR WORK

The idea of constructing a deep neural network as a composition of reusable computation blocks has been extensively explored in recent literature. First conditioned computation methods used evolution algorithms (Wierstra et al. (2005); Floreano et al. (2008); Miikkulainen et al. (2019)) to determine a suitable model architecture; later on, reinforcement learning was used to optimize the model layout and parameters (Zoph & Le (2016); Bengio (2016); Baker et al. (2016)). Recently, Kirsch et al. (2018a) proposed an end-to-end modular neural network framework that learns to choose the best among several module candidates at each layer according to the input data.

Conditional computation routing is even more appealing for the multitask learning application, as it allows to both learn reusable computation blocks and adjust the model to each specific task with minimal network alteration. For instance, Misra et al. (2016) use an additional set of modules for each task and enforce similarity between the corresponding task-specific modules weights. Rosenbaum et al. (2017; 2019) used reinforcement learning to perform task- and data-specific routing; Sun et al. (2019) train both the model weights and the task-specific policy that determines which layers should be shared for a given task; Ma et al. (2019); Maziarz et al. (2019) introduced a multi-task model where each layer's output is computed as a weighted sum of a set or candidate modules outputs, which is similar to our method, with two important differences: a) in the works by Ma et al. (2019); Maziarz et al. (2019), the output is computed as a linear combination of candidates outputs, not module weights, and b) the module candidates in their methods are tied to the specific location in the network. Maninis et al. (2019) added a task-specific residual adapters to specialize the feature extractor to each task, The method proposed by Purushwalkam et al. (2019) performs zero-shot multitask learning by finding a task-specific routing using a gating mechanism. This approach is somewhat similar to ours, but is less efficient in terms of the number of parameters since a) the method in Purushwalkam et al. (2019) uses a modular network on top of a classical ResNet (He et al. (2016)) while our method uses only about 100 additional parameters and b) the modules used in Purushwalkam et al. (2019) are layer-specific and not reusable while there is no such constraint in our architecture. Wu et al. (2018); Newell et al. (2019); Guo et al. (2020) perform task-specific model compression. There also exists a number of studies on the effectiveness of modular networks in the context of visual question answering. One of the early papers Andreas et al. (2016) used a natural language parser to determine the layout of the composite network consisting of predefined modules that solve different kinds of subtasks, such as find, measure, describe etc; further works improved this approach by switching from an external parser to a fixed (Hu et al. (2017b)) or arbitrary query structure (Hu et al. (2017a; 2018); Pahuja et al. (2019)).

## 3 COMPOSITIONAL MODELS AND MODULE MIXTURES

### 3.1 COMPOSITIONAL MODELS

In conventional convolutional deep neural networks, most layers are different from each other and do not conform to a particular fixed design. Various model blocks may have different resolutions and different numbers of filters, some blocks may have or lack a residual connection and so on. This makes most model activations at different layers incompatible with each other. Recently, it has been demonstrated (Sandler et al., 2019) that high performing convolutional networks can nevertheless be composed of identical blocks. Such networks that were called *isometric networks* essentially iterate on the same activation space. The architecture of an isometric network includes the following core components: **(a)** *input adapter* that maps model input $i \in I$ into the activation space $Z$; **(b)** *model body*, a sequence of *blocks* all sharing the same architecture and mapping the space $Z$ to itself; **(c)** *output adapter* mapping the output of the last block into the embedding space $E$, and finally **(d)** *logits layer* for classification models, mapping embedding space $E$ into the final predicted probabilities.

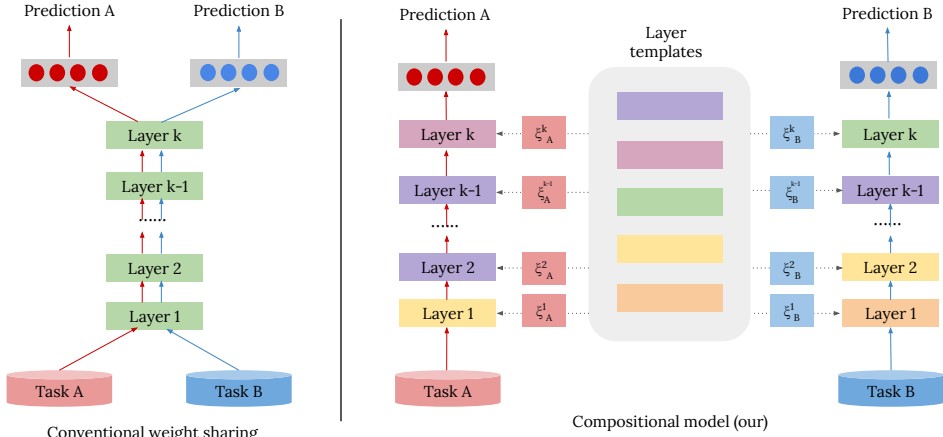

Figure 1: **Left:** conventional hard weight sharing approach. The model consists of a shared feature extractor followed by the task-specific task solvers (e.g. logits layer in case of classification). The only place where the model can specialize to each task is in its solver, which might not be enough for sufficiently distinct tasks. **Right:** Compositional model. We use a shared set of trainable modules (we call them *templates*) available for all tasks; each layer's weights are generated as a linear combination of the templates weights, and the linear combination weights (*mixture weights $\xi$*) are task-specific. In this case, the templates can not only be shared across different tasks, they also can be reused multiple times within the same task. (best viewed in color).

While isometric networks achieve near SOTA performance, the fact that they iterate on the same internal activation space $Z$ makes them a perfect candidate for studying *modular architectures*. Specifically, instead of treating each block as an independently trainable mapping, we may view the entire model body as a composition of modules $f_k$ each of which may appear in a sequence more than once. In the following, we will denote a set of trainable modules as $\mathcal{F} = \{f_1, f_2, \ldots, f_K\}$ and their composition $f = f_{m_L} \circ \cdots \circ f_{m_1}$ forming a model body[1] will frequently be denoted as $f_{(m_1,\ldots,m_L)}$ with $(m_1, \ldots, m_L)$ being called a *signature* of this model.

### 3.2 Learning Compositional Models

**Mixture modules** Compositional models discussed in Section 3.1 can be trained on a particular task for any predefined choice of the model signature. But, while the study of hand-designed modular architectures is an interesting exploration in itself, one may want to go further and identify a sequence of modules $(m_1, m_2, \ldots, m_L)$ together with the modules $\mathcal{F}$ themselves that is the most advantageous choice for solving a particular problem. In some previous explorations of modular architectures, authors used a probabilistic approach where the model body $(m_1, \ldots, m_L)$ was sampled for every input batch or sample from the optimized probability distribution $p(m_1, m_2, \ldots, m_L)$. For example, in Kirsch et al. (2018b), modules[2] were sampled from a distribution defined by per-layer conditional probabilities $p(m_\ell | x_{\ell-1})$ with $x_{\ell-1}$ being the output of the previous layer $(\ell - 1)$, i.e., $x_{\ell-1} = f_{(m_1,\ldots,m_{\ell-1})}(i)$. Since discrete nature of such a choice limits the flexibility of the model and since in practice learning such a model may be complicated due to high-variance estimator noise, we instead replace a composition of blocks from $\mathcal{F}$ with a composition of *mixture modules*, each of which is essentially a smooth superposition of blocks from $\mathcal{F}$. More precisely, each mixture module $h_{\boldsymbol{\xi}}$ with $\boldsymbol{\xi} \in \mathbb{R}^K$ has the same architecture as blocks from $\mathcal{F}$, but every parameter $\theta$ in this mixture module (be that some convolutional kernel, or a batch normalization $\beta$ or $\gamma$ parameter) is a linear combination $\sum_{k=1}^{K} \xi_k \theta_k$ of the corresponding parameters $\theta_k$ from $f_k \in \mathcal{F}$. Notice that since each block contains multiple convolutions with different nonlinearities, a mixture module obtained by linearly combining weights from $\mathcal{F}$ is not a linear combination of modules from $\mathcal{F}$. It is true however that if $\xi_i = \delta_{ij}$ for some fixed[3] $j$ (which we later refer to as "one-hot" vectors), a mixture module $h_\xi$ becomes identical to $f_j \in \mathcal{F}$. Due to the fact that $h_\xi$ mixes parameters from $\mathcal{F}$, we will

---

[1] with $L$ is the total number of layers in the model

[2] here each $m_k$ is actually a set of several chosen modules the output of which is combined

[3] Here $\delta_{ij}$ is a Kronecker delta

sometimes refer to blocks from $\mathcal{F}$ as *templates*, which are mixed together to finally form an actual unit of computation.

After we choose a particular task and replace the model body with a composition of mixture modules $h_{\boldsymbol{\xi}_L} \circ \cdots \circ h_{\boldsymbol{\xi}_1}$, we obtain a differentiable objective that can be optimized with respect to a *model signature* $\Xi = (\boldsymbol{\xi}_1, \ldots, \boldsymbol{\xi}_L)$ and parameters of $\mathcal{F}$. In our experiments, we constrained each of $\xi_\ell$ by using a softmax nonlinearity, which allowed us to guarantee that $\sum_i (\xi_\ell)_i = 1$ for every layer $\ell$. Of course, even with this choice of the nonlinearity, we are not guaranteed that the final signature $\Xi$ will contain only one-hot vectors $\xi_\ell$ that can be interpreted as blocks from $\mathcal{F}$.

**Regularization of mixture weights**  While in our experiments, mixture weights $\xi_\ell$ frequently approached a set of approximately one-hot vectors[4], one can use additional regularizers to ensure that all $\xi_\ell$ are one-hot and the resulting model can be represented as a composition of individual template modules. In Appendix A, we outline two choices that we experimented with: (a) entropy regularization that penalizes $\max_i (\xi_\ell)_i < 1$, but does not involve template weights and (b) a regularizer that penalizes $\Xi$ and also effectively attracts individual templates towards the generated mixture weights to which they contribute the most.

**Weight generation interpretation**  Consider a single block of our isometric network and let $\mathfrak{F}$ be a function family containing single-block tensor transformations corresponding to all possible choices of block weights. All neural networks obtained by composing $L$ such blocks without any weight-tying form a function family $\mathfrak{F}^L$. Now, instead, consider a model composed of mixture-weight blocks with templates from $\mathcal{F}$. In this case, the family of all possible single-block transformations is only controlled by a vector of mixture weights $\boldsymbol{\xi}$ and narrows down to a family further denoted as $\mathfrak{F}_{\mathcal{F}}$. A composition of mixture-weights blocks is thus $\mathfrak{F}_{\mathcal{F}}^L$. In a sense, choosing templates $\mathcal{F}$, we introduce a small function family that can be parameterized by just a handful of real-valued parameters $\Xi$. This perspective shows a connection of the proposed approach to other approaches parameterizing neural network weights like, for example, HYPERNETWORKS (Ha et al., 2017).

## 4 APPLICATIONS AND EXPERIMENTAL RESULTS

### 4.1 ISOMETRIC MODEL DETAILS

Before discussing our experimental results, we first need to briefly describe the architecture of the Isometric network used as a basis of these experiments. As described in Section 3.1, our implementation of the modular Isometric network contains the following four components (see Appendix B.1 for details): **(a)** *input adapter*, a nonlinear convolution with the kernel and the stride of the same size (typically 4 to 16), which effectively downsamples the image while increasing the number of channels in the tensor to $40\varrho$, where $\varrho$ is the *model depth multiplier* controlling the complexity of the network; **(b)** *model block*, a typical MobileNetV3 (Howard et al., 2019), or Isometric model residual block (Sandler et al., 2019) with the "expansion", depthwise and "projection" convolutions and a squeeze-and-excite block (Hu et al., 2020); **(c)** *output adapter*, a sequence of two nonlinear kernel-1 convolutions with an average pooling layer in between and finally **(d)** *logits layer*, a fully-connected layer producing model predictions. In the following, we will rely on two specific designs: (a) larger design more suitable for complex tasks that uses 40-channel tensors (or $\varrho = 1$) and the embedding size of 1280 (as produced by the output adapter) and (b) smaller design typically using 24-channel tensors[5] and the embedding size of 256.

In our implementation of networks with mixture modules, we used mixture weights $\xi \in \mathbb{R}^K$ for mixing all variables appearing in every component of the block except for batch normalization (BN) parameters: this included all convolutional weights and biases, and all components of the squeeze-and-excite block. In some experiments, we also used the same $\xi$ to produce a linear combination of batch normalization $\beta$ and $\gamma$ parameters (but never moving mean variables). In the following, we will refer to such experiments as experiments with "mixed batch normalization parameters".

Notice that generating block parameters from templates and mixture weights can be expensive for a large number of templates in $\mathcal{F}$. However, since in this work we perform this calculation for an entire

---

[4]i.e., a state with $\max_i (\xi_\ell)_i \approx 1$ for most layers $\ell$

[5]We used the number of channels divisible by 8 for training efficiency.

Table 1: IMAGENET results, comparing performance of different module sequences. We use 8 and 16-layer isometric models from Sandler et al. (2019). For our method we use the same models, but the modules are arranged into 32 and 48 layers as described in Section 4.2. The models are of the same size as baseline.

| Isometric-8 | 8/16 model | Isometric-16 | 16/48 model |
|---|---|---|---|
| 68.3 | **70.6 (+2.3%)** | 71.7 | **72.8 (+1.1%)** |

batch and not for individual samples, this procedure did not introduce a significant computational overhead.

## 4.2 LEARNING A SINGLE TASK

We began our study of networks with mixture weights by training these models on individual supervised tasks. In our initial experiments, we studied models with 4 to 32 layers and 2 to 16 templates forming $\mathcal{F}$. We trained these models on the CIFAR-100 dataset using a 24-channel tensors and a smaller output adapter (see Appendix B.1).

When training a model on a single task, we observe that even in the absence of regularization, the mixture weights $\Xi$ frequently approach a collection of one-hot vectors, i.e., $\max_i(\xi_\ell)_i \approx 1$ for most layers $\ell$. Discovered model signatures (with one regularized example having a signature $(1, 2, 2, 3, 3, 3, 3, 4, 4, 4, 4, 4)$) are discussed in Appendix C in detail. After studying multiple such signatures, we observed that they tend to repeat the same modules in contiguous uninterrupted sequences and the number of repetitions grows towards the end of the model. In other words, early modules use fewer repetitions. We call such pattern *progressive*.

Once the optimal signature is identified, we investigated whether this particular model architecture is truly optimal for solving a particular task. We compared networks with mixture weights to those with hand-designed signatures of three types: **(a)** *sequential signatures* $(f_K^{L/K}) \circ \cdots \circ (f_1^{L/K})$ repeating each of $K$ templates $L/K$ times, where $L$ is the number of layers; **(b)** *cyclic signatures* corresponding to $(f_K \circ \cdots \circ f_1)^{L/K}$ and **(c)** random signatures. Our experimental results (for details see Appendix C) appear to indicate that mixture-weight results are comparable to those obtained with a sequential signature and overall outperform results for other considered model signatures. However, it remains an open question whether progressive signatures are actually optimal, or are an artifact of the modular network training.

In Table 1 we show our results of improving IMAGENET accuracy by increasing the number of layers while keeping the number of parameters intact. Due to hardware and time limitations we only experiment with progressive pattern and leave more detailed study for future work. More details on IMAGENET architecture is included in the Appendix.

## 4.3 MULTI-TASK AND TRANSFER LEARNING

**Multi-task learning** Another potential application of compositional models is in multi-task learning, where we expect the computation to be partially shared across different tasks. In this application, each task $T_\nu$ uses its own private set of mixture weights $\Xi_\nu$, but the template blocks $\mathcal{F}$ and input, output adapters are shared across all tasks. As the number of tasks grows, the total number of blocks in $\mathcal{F}$ should also be scaled up to achieve sufficiently high accuracy on all $\{T_\nu\}$. Also note that modular networks provide a convenient tool for training sets of tasks parameterized by additional arguments like those, for example, used in basic visual question answering modules (finding specific objects, transforming and filtering attention regions, etc.) (Hu et al., 2018; 2017a).

Notice that in its current form, this approach is a generalization of ADASHARE (Sun et al., 2019), in which reused modules always retain their relative order with respect to each other[6]. Since multi-task training is not the primary application explored in this paper and the direct comparison is not possible without re-implementing the method, we do not compare our results with ADASHARE.

While the tasks $\{T_\nu\}$ could be very diverse and along with conventional supervised tasks could also include unsupervised tasks, or even generative and image-to-image translation objectives, here we

---

[6]and cannot be reused later in the same network

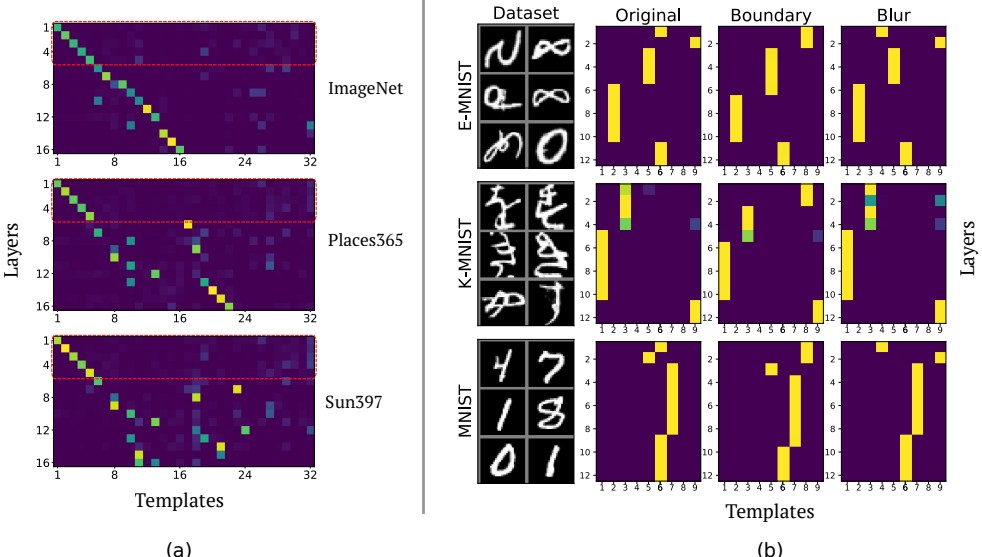

Figure 2: Self-organization of modules for multi-task and domain-adaption. In (a) co-training 5 natural image datasets with 32 modules (CIFAR-100 and FOOD101 not shown), first 5 layers were reused across all learned models (shown with dashed rectangles). In (b) co-training 9 modules on 9 tasks: 3 handwritten digit datasets EMNIST, MNIST and KMNIST with the original images and two image augmentations (boundary extraction and blurring). The model learned to use different input transformation for each image augmentation, while sharing the top layers for each task. On the other hand, for each fixed augmentation, the model learned to share first few layers across all different tasks.

will only consider a collection of image classification tasks. For our experiments, we chose a collection of natural image datasets including: IMAGENET, CIFAR-100, PLACES365, SUN397 and FOOD101. Here we used model depth multiplier $\varrho$ of 1 and a larger output adapter (see Section 4.1). We compared multiple different ways of co-training different classification tasks: **(a)** baseline model with or without per-task batch normalization $\beta$ and $\gamma$ variables (see *linear patches* in Mudrakarta et al. (2019)) and **(b)** model with mixture-weights with or without per-task batch normalization variables.

In our experiments summarized in Appendix D, we noticed that the mixture-weight model with $K = 32$ templates achieved a lower overall cross-entropy loss and thus fitted training data better than the baseline models. This did translate to a superior IMAGENET performance, but in some cases resulted in worse accuracies for simpler datasets, on which overfitting seemed to occur.

**Self-Organization of Computational Blocks**  Perhaps one the most interesting aspects of networks trained with mixture modules is that they essentially involve self-organization of atomic computational blocks. The properties of emergent block configurations are insightful both when we train a model on a single task and when performing multi-task learning.

In the case of a multi-task learning, we discover that the location of blocks re-used across tasks agrees with our intuition regarding the nature of the task similarity. Specifically, if all tasks operate on a similar domain, first few layers of all models appear to share the same mixture weights (see Figure 2(a)) thus sharing early-stage image processing. For similar tasks operating on different image domains, we see the opposite situation (see Figure 2(b)), where the first few layers differ, but the late sample processing is shared across all tasks performing inference on the same dataset (EMNIST, MNIST and KMNIST in our experiments). In this case, first few layers essentially adapt network input to a particular common format shared across all related tasks and processed by the following layers.

Table 2: Transfer learning results from model trained on multiple initial datasets (IMAGENET, CIFAR-100, PLACES365, SUN397 and FOOD101) onto new datasets. For baseline column we train multi-task model following Mudrakarta et al. (2019), with shared backbone and per-task logits and batch normalization parameters. We then transfer to the target datasets by fine-tuning logits and batch normalization parameters. Other columns show results obtained by our method, where we use equivalent architecture, but with mixture weights both for initial model and each transfer task. Each column shows results with mixture coefficients initialized from the corresponding training task. Our method adds less than 500 additional parameters for each transfer task.

| Dataset | Baseline | IMAGENET | CIFAR | FOOD |
|---|---|---|---|---|
| AIRCRAFT | 43.3 | **43.9** | 42.6 | 39.4 |
| CARS | 49.2 | **51.0** | 46.7 | 42.2 |
| EMNIST | **85.5** | 85.0 | 85.2 | 83.6 |
| STANFORD DOGS | 55.6 | **57.6** | 42.4 | 36.5 |

**Transfer learning**   A set of blocks $\mathcal{F}$ pretrained on a collection of tasks can later be used for transferring extracted knowledge onto a new task. Adjusting mixture weights to form a new model potentially gives us more flexibility than tuning logits or batch normalization parameters alone.

In our experiments, we used models pretrained on 5 datasets: IMAGENET, CIFAR-100, PLACES365, SUN397 and FOOD101 (see Appendix D). We then used a learned collection of templates $\mathcal{F}$ for transferring knowledge to different new tasks such as STANFORD DOGS, AIRCRAFT, CARS196 and EMNIST. We compared our results with the baseline obtained by fine-tuning logits and BN parameters of a baseline model trained with a common model backbone, but per-task logit layers and BN parameters ("Baseline with patches" in Table 7). Proper initialization of mixture weights $\Xi$ proved to be extremely important for achieving best performance. Model initialized with random $\Xi$ performed significantly worse than the models initialized with $\Xi$ corresponding to one of the datasets in the original mixture. In the following, we will denote by $\Xi_\nu$ a set of mixture weights used for a task $\nu$ in the original mixture-module network trained on a collection of datasets. Initializing the model at some particular $\Xi_\nu$, we compared model performance with and without fine-tuning $\Xi$ (fine-tuning BN parameters and logits in both cases).

The final results are presented in Table 2. Experimental observations suggest that a proper choice of the dataset $\nu$ for $\Xi = \Xi_\nu$ is extremely important: using $\Xi$ values from the FOOD196 model resulted in a model that achieved the same $> 99\%$ training accuracy, but would frequently lead to a $10\% - 20\%$ smaller validation accuracy. We also observe that the validation accuracy improvement in the final model can be partly attributed to a higher validation accuracy on the chosen model $\Xi_\nu$ and partly to the result of fine-tuning this $\Xi \approx \Xi_\nu$ value.

## 4.4   DOMAIN ADAPTATION

In this section, we explore the use of compositional models for unsupervised multi-source domain adaptation which aims to minimize the domain shift between multiple labeled source domains and an unlabeled target domain. To our knowledge, we are the first to show that it is possible to perform domain adaptation with modular networks by simply changing the order of modules in the network. To illustrate the ability of compositional networks to perform distribution alignment, we adopted two existing unsupervised domain adaptation methods: Adversarial Discriminative Domain Adaptation (ADDA) by Tzeng et al. (2017) and Moment Matching for Multi-Source Domain Adaptation (M3SDA) by Peng et al. (2019). We performed experiments with two sets of datasets: digits datasets (MNIST, corrupted MNIST, SVHN and USPS) and DomainNet dataset introduced by Peng et al. (2019). The code used for the domain adaptation experiments can be found on our project page (*to be added if accepted*). The details of implementation of the compositional domain adaptation and a review of related work on domain adaptation can be found in the Appendix E.

**Multi-source adversarial classification**   In the compositional adversarial classification method, the shared classifier is trained to perform accurate classification on all source datasets. Additionally, a domain discriminator is trained to distinguish between the late features of source and target domains (in our implementation, the output of the layer before the output adapter). As in the multitask learning scenario, the model contains separate mixture weights for each domain; the mixture weights

Table 3: Multi-source domain adaptation target test accuracy results with compositional models on Digits domains (MNIST, corrupted MNIST (shear), corrupted MNIST (scale), corrupted MNIST (shot noise), SVHN → USPS) and DomainNet domains (clipart, infograph, quickdraw, painting, real → sketch). Baseline methods: Adversarial Discriminative method and Multi-Source Moment Matching method. We pretrained the baseline models on source domains and fine-tuned only the input adapter (IA) containing ≈ 1000 parameters. We applied the same domain adaptation techniques for our conversational model and fine-tuned only mixture weights (MW) consisting of 64 parameters.

| Method | # parameters | Digits | DomainNet |
|---|---|---|---|
| Source-only | − | 85.8 | 13.6 |
| Adversarial | 680 | 93.4 | 14.5 |
| Compositional Adversarial (our) | 64 | 92.2 | 13.8 |
| Moment Matching | 680 | 94.2 | **15.6** |
| Compositional Moment Matching (our) | 64 | **94.4** | 14.4 |

of source domains are trained along with templates to minimize the classification loss, while the target domain mixture weights are trained to "fool" the domain discriminator by aligning the target feature with the source features.

**Moment Matching**    As in the original paper introduced by Peng et al. (2019), the method performs domain adaptation by minimizing the distance between the source and target domain statistics (more details can be found in the original paper). In the compositional moment matching setting, only target domain mixture weights are trained to perform moment matching. In addition to that, the model has two logits layer instead of one which are used to perform minmax discrepancy optimization (see Peng et al. (2019) for more detail). In the compositional case, only the mixture weights are affected by the discrepancy minimization.

**Results**    As we can see in the Table 3, fine-tuning only the mixture weights for the domain adaptation objective improves the classification accuracy on target domain. To compare the results of conversational domain adaptation, we trained the adversarial and moment matching models with an isometric model without weight sharing as a backbone, and for fair comparison we only fine-tuned the input adapter (Saito et al. (2019) point out that finetuning the first layers is appropriate for the case when the domain shift is mostly in the low-level visual features). The results show that although only fine-tuning mixture weights of the conversational model does not result in the same adaptation efficiency as the baseline probably due to the fact that mixture weights contain more than ten times parameters than the input adapter, such approach can still significantly increase the target accuracy.

## 5    DISCUSSION

In this paper, we introduced a novel approach to modular neural networks that decouples modules from their position in the network thus allowing reuse of modules not only across tasks, but also across layers. Our method beats the baseline on multitask learning with additional templates and achieves better transfer learning results on most of the datasets we explored, which we partly attribute to less aggressive parameter over-sharing and partly to the ability of our model to fine-tune the module order itself. We then apply our method to training parameter-efficient networks and show that we can scale SOTA IMAGENET classification models to achieve higher accuracy while keeping the number of parameters fixed. We also show that simple fine-tuning of mixture weights allows unsupervised domain adaptation improving model's performance on target domain by a large margin.

Since, by design, the compositional models divides the task into simple subtasks, it can provide greater understanding of the computation process. For example, we notice that co-training multiple tasks on the same image domain, all trained models share early image processing modules. At the same time, when the dataset contains multiple image domains of the otherwise identical task, early layers bring network activations to the common format, while the final stage of the computation defined by the sequence of participating modules is shared.

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
