# OpenReview forum: "Compositional Models: Multi-Task Learning and Knowledge Transfer with Modular Networks"
_ICLR.cc/2021/Conference — Reject_

### Official Review · AnonReviewer1 · 2020-10-21

**Rating:** 4
**Confidence:** 4

**Review:**

==========

Summary:
This paper proposes a modular neural network architecture that consists of module templates that can be invoked repeatedly in different layers and combined using mixture weights.
It can be trained end-to-end by linearly combining parameters from different modules. It shows promising results on multiple applications such as transfer learning and domain adaptation

==========

Strength:

- Modular architecture allows parameter efficient and interpretable models and I believe this is an important direction to explore for many applications, including the applications considered in this paper.

- Proposes a general formulation where module templates can be reused multiple times in different layers and can be trained end-to-end by softly combining module parameters.

- Self-organization of modules and how they are reused, as shown in Figure 2, is interesting and can be used to interpret what each module is learning.

- Method is simple, clear, and well written.

==========

Weakeness:

- Sec 4.2 & appendix C. Quantitative results do not sufficiently show if the proposed method is actually better. The accuracies are pretty close to each other and the numbers are from a single run so they are not statistically convincing.

- Sec 4.3 multi-task learning & appendix D, Section 4.4. same as above.

- Table 2. It says results are also from Places365 and sun397, but they are not shown in the table

- Section 4.3. It requires a manual choice of initial template weights that is learned from some dataset, and otherwise, it can result in worse performance. This is a strong limiting factor and if the model is able to utilize the template modules, it should be able to learn the weights.

==========

This paper proposes an interesting way of composing neural modules for a number of applications, but the experimental results do not sufficiently demonstrate the advantage of using the proposed approach.
I believe more exploration in this direction and making the results concrete by showing how this model can be better than baseline models would make the paper much stronger.

---

> ### Author Response · Authors · 2020-11-17
> **Reply to Reviewer 1**
>
> We would like to thank Reviewer 1 for their thorough and thoughtful comments. Here we would like to address some of the questions raised by the reviewer:
>
> * We agree that the empirical evidence we accumulated so far is not entirely convincing, especially in the light of not gathering enough statistics to get even a rough idea of confidence intervals of our measurements. We will address this issue in the future version of the paper.
> * Results shown in Table 2 used the multi-task model trained on 5 datasets including Places365 and sun397, however, when doing fine-tuning, we only conducted experiments using pre-trained mixture weights from 3 models. While preliminary results for mixture weights taken from Places365 and sun397 do not provide any insight and are comparable to those obtained for food101 and cifar100, we agree that they need to be included for completeness.
> * It is true that the manual choice of initial template weights used in Section 4.3 can be very limiting for the model performance. In the future version of the paper, we will also introduce and discuss a superior method that does not rely on this manual choice and  compare it to a previously used manual choice of mixture weights.

---

### Official Review · AnonReviewer3 · 2020-10-27
**very interesting behavior, but strays from focus**

**Rating:** 5
**Confidence:** 4

**Review:**

This paper presents a method of composing stacked neural network blocks by linearly combining module "template" weights.  The work extends "isometric networks", in which each block in the stack has the same operational structure, by parameterizing each block using a mixture of the weights from a bank of K blocks ("templates").  In multitask learning experiments, the mixtures naturally learn to share common components between tasks, while learning task-specific components where needed.  Further experiments describe behavior of the system applied to transfer learning and domain adaptation scenarios.

The most interesting findings are in the naturally arising patterns of template sharing:  When these occur, and why, including the occurrence of the "progressive" pattern of repeated blocks.  I feel this can be even further developed, and tied to existing architectures in the discussion -- for example, resnet architectures are progressive, and this model learns progressive blocks on its own:  this suggests the model converged to a similar structure, reinforcing the importance of this trait in classification deep nets.

The other two applications, transfer learning and domain adaptation, seem underdeveloped.  The transfer learning experiment compares only against a baseline model with similarly constructed initialization but fewer transfer-learned parameters, so it's hard to know what to take away from this (see also comments and questions below).  Even though the mixture weights are relatively few parameters, it's not clear that these are usefully fewer in most settings:  full

The main multitask investigation has much interesting discussion relegated to supplementary materials.  The appendices should be in the main PDF (I almost didn't see them in the supplemental zip file; particularly because they are referred to as "appendices" but were not appended).  I also think some of this material, particularly much of appendices C and D, would be better suited to the main text in order to flesh out the multitask behavior.

Overall I find the approach and the investigation of multitask training very interesting.  However, the text also lacks clear descriptions of many of the setups and comparisons, and sacrifices a deeper dive into the multitask behavior and ties to existing architectures, in order to include additional tasks that do not have clear takeaways.




Additional comments and questions:

- at inference time, it seems one could construct model weights \theta using the mixture and use these directly, discarding the templates.  This would probably be more efficient for most scenarios; it would be good add a comment about this, I don't see it explicitly mentioned.

- p.3: what is the definition of m_l ?

- fig 2a:  how are the columns sorted ?

- fig 2b:  I slightly disagree with the last sentence of the caption, "for each augmentation ... learned to share first few layers across all different tasks" --- in fact, this looks to be the case only for the "boundary" augmentation, not each of the augmentations.  Nevertheless, the fact that it happened in this case seems significant, and the claim could be revised to e.g. "in the case of the boundary augmentation ...".

- how quickly do the mixtures converge to mostly 1-hot, how do they evolve over the course of training?

- table 1:  I don't understand what are the "8/16" and "16/48" models --- what do the 8 and 16 in "8/16" correspond to?  sec 4.2 doesn't have this notation

- fig 2:  why are cifar10 and food101 not shown?  they would be interesting to see, e.g. if first layers are different due to different resolution, for example

- some mixtures, especially learned via sgd, can suffer from partial collapse where just one or a few components that happen to perform slightly better at the start of training "take over" and starve out potential use of remaining components, due to a self-reinforcing loop of component model weights being updated to perform better, leading to its mixture coefficient increasing, which leads to further model improvement, larger coefficient, etc.  Have you seen this model suffer from a problem like this?

- For transfer learning experiment:  This can have more comparisons.  I think the baseline uses the same initial model (using \xi_\nu) as imagenet, and fine-tunes just the batchnorm and last logits layers on the new dataset.  This one point of comparison, but I'm not convinced it is the strongest baseline --- one could also train all model weights, for example.

- Are there any benefits to constraining weight updates to just the mixture coefficients \xi compared to training all weights?  Yet another strategy could be to initialize model weights using the best initial \xi, but instead of learning a new mixture, construct the model with mixed weights \theta = sum(\xi_i \theta_i), and fine-tune the model weights \theta of each layer directly on the new dataset (eliminating the mixture entirely at this point).

- Are there any transfer datasets for which a different initial model choice performs better than imagenet?  What about cifar10, is it better using imagenet or cifar100?


- Domain adaptation experiment:  I find this experiment very interesting, that such similar performance can be obtained just by tuning the mixtures.  I think it could be developed further:  Are the templates that are changed more at the first layers or at all layers?  And how does this compare to learning all weights instead of just the mixture?

---

> ### Author Response · Authors · 2020-11-17
> **Reply to Reviewer 3**
>
> We would like to thank Reviewer 3 for their thorough and thoughtful comments. Here we would like to address some of the questions raised by the reviewer:
>
> * **Multitask learning: more comparisons ... needed.** Indeed, we agree that comparison with other baseline multitask learning methods are necessary to draw conclusions on the efficiency of our method. We will make sure to include them.
> * **... one could construct model weights ... discarding the templates.** This is indeed true and we will add a comment clarifying this point in the future version of the paper.
> * **p.3: what is the definition of m_l?** In Section 3.1, $(m_1,\dots,m_L)$ is a tuple of integer numbers in the range $1,\dots,K$. Here they are assumed to be chosen and fixed. Later in Section 3.2 we discuss prior work treating $m_k$ as random variables sampled from a learned distribution and transition towards mixture weights.
> * **Some mixtures ... can suffer from partial collapse … Have you seen ... a problem like this?** Interestingly, we virtually never observed this effect. In the multi-task case, we only occasionally saw one template (out of 32 or more) hardly being used by any of the models. This effect indeed deserves a more thorough discussion in the paper and we will add a more in-depth study of this phenomenon.
> * **Fig 2b: I slightly disagree with the last sentence of the caption.** This is a great point. This is a wording mistake and we will correct this caption.
> * **Fig 2a: how are the columns sorted?** We sorted the columns following a simple qualitative heuristic: the templates with mixture weights over the threshold used in ImageNet are sorted in the ascending order with respect to the layer they appear in, same is then done for the remaining templates in places365, then sun397, food101 / cifar100 (not shown). This sorting method is only used for visualization and does not affect any other results.
> * **Transfer learning experiment**. This can have more comparisons. We agree with this point and will try to address this in the future version of the manuscript.
> * **Table 1**: This notation is indeed confusing and we will address this in the future version of the paper. Here it means that we use 8 templates with 16 layers and 16 templates with 48 layers correspondingly.
> * **Fig 2: why are cifar100 and food101 not shown?** We did not include these results only because of the publication size constraint. We observed that for these datasets the first 5 layers were in fact the same as in all of the other datasets. We will include these in the future version of the paper.
> * **Are there any benefits to constraining weight updates to just the mixture coefficients compared to training all weights?** Fine-tuning only mixture weights might be beneficial in the case when we want to adapt many pretrained templates to new tasks without a need to store the weights for all of them. In this case, when we only fine-tune mixture weights, we need to store only T*L parameters, where T is the number of templates and L is the number of layers. In one potential application, one could maintain a centralized databank of "frozen" templates, which can grow when necessary and models trained on different new datasets become compact "mixture weight" records stored alongside templates.
> * **Are there any transfer datasets for which a different initial model choice performs better than imagenet? What about cifar10, is it better using imagenet or cifar100?** In our experiments, we observe that for a new task the best choice of initial mixture weights is usually the mixture weights from the closest source dataset - closest semantically and visually. In particular, the best choice for the initial MWs for CIFAR10 is indeed the MWs from CIFAR100. We realize that the sensitivity of our model to MW initialization is a serious issue that we address in our current studies.
> * **Domain Adaptation experiments: Are the templates that are changed more at the first layers or at all layers?** Thank you for raising these very interesting questions. We will make sure to illustrate the examples of the change in MWs after adapting the model to the target domain in the next version of the paper. As for training all parameters, we did not include these results since in this setting the behavior of the compositional model should not differ much from the standard DA methods. The practical reason for fine-tuning only mixture weights is the parameter-efficient model personalization: imagine we have a large-scale model pre-trained on a set of domains, and we would like to adjust it for thousands of users each of which is a separate target domain. In case of fine-tuning of mixture weights, we would only need to store around an order of hundred parameters; in the standard case when we fine-tune the whole network, we would need to store a separate copy of the model for each domain which is unfeasible in a large-scale application.

---

### Official Review · AnonReviewer4 · 2020-10-28
**Exciting research direction but substantial modifications required**

**Rating:** 4
**Confidence:** 4

**Review:**

The authors propose a method to design network architectures as a combination of network components. Their idea consists in imposing an architecture that is a sequence of identical network blocks, and learn a series of templates, which provide an instantiation of model weights for one such network block. Model weights are then estimated for a specific task as a linear combination of a set of template weights.


STRENGTHS

The idea of using modular networks to share knowledge across tasks and domain is appealing, as is the potential interpretability with regards to which templates are preferred.
The concept of weight templates is interesting, as it affords to share knowledge implicitly, and provides a flexible setting for network construction.
Authors evaluate their approach on a very large set of problems and datasets, and provide promising performance. The experiment studying similarities across network signature with respect to the problem setting is particularly interesting.
The domain adaptation setting, with a significantly lower number of parameters is additionally quite promising.


WEAKNESSES

-One of the main limitations of the work is the unclear or incremental novelty with respect to other works performing soft weight sharing, and that implement modular networks techniques. Differences are not clearly stated, in particular with respect to the works of Ma et al, Maziarz et al (linear combination of module candidates), Kirsch et al (probabilistic sampling of modules), and Sun et al (shared modules for domain adaptation). A clear list of contributions as well as a dedicated section analysing main innovations with respect to these highly similar works would be very beneficial.
In particular, a claimed innovation/main difference in section 3.2 is the use of a mixture of templates to compute the weights of a specific network layer. However in practice, mixture weights are regularised to be closest to one-hot encodings, leading to a composition of blocks rather than a composition of mixtures of blocks.  Can authors please comment on this point?

-The regularisation strategy is an important aspect of the model and its performance. Details regarding the adopted method and proposed solution should be present in the main method description, and the two strategies adopted (vs no regularisation) should be evaluated in a dedicated ablation sections. Can authors comment on whether templates are regularised to comprise different sets of weights, and whether the problem of templates being too similar arises in practice?

-Authors have made a lot of effort towards evaluating their approach in different scenarios and settings. Unfortunately, the experimental evaluation section needs substantial modifications to provide thorough and reproducible evaluation of the work.
The first issue relates to the first two set of experiments, (single task and multi-task) for which only vague descriptions are provided: experimental setting, datasets, network architectures (number of layers, number of templates)…Most of these important details are either absent from the paper or described only in the supplementary material. This makes it very challenging to understand exactly the experimental setting, and further more concerning is the fact that results are only provided and discussed in the supplementary material. In particular, while the datasets used are, for the most part, very common, providing a description in the supplementary material is important for reproducibility (regardless of code sharing).
The issue is present throughout the whole experimental section, with key details missing from the experiment descriptions (notably number of templates vs number of layers), and some important details are only provided in table captions.

-In addition, the evaluation is missing important comparisons to closest works. Problems like knowledge transfer and multi-task learning are evaluated using novel settings that differ from standard benchmarks in the fields, preventing from providing context and directly comparing to similar approaches. For example, the multi-task learning section claims unavailability to compare to similar work Adashare, despite the fact that this work has been evaluated on standard, publicly available benchmarks. Can the authors justify this decision? Why not use standard benchmarks to evaluate the proposed method?

-The domain adaptation/transfer learning sections are interesting, providing promising results. However, wouldn’t it be more advantageous to learn templates across multiple datasets, which would allow to extract common knowledge and facilitate transfer to a new dataset/domain?

RECOMMENDATION

In summary, the idea of learning modular, compositional architectures, with a set of pre-trained templates that can be reused in different problem settings is quite exciting.
However, the paper in its current form suffer from a set of important flaws, and would need substantial modifications. I would recommend 2 key changes:

1-	A clarification of novelty/contributions, in particular with respect to pre-existing modular architecture works and soft sharing methods. I would also recommend focusing a little more on the regularisation strategy.
2-	Revamping the evaluation section in a significant way: a) first by providing a setting where the approach can be compared to similar works on a standard benchmark, b) providing a clear description of a set of experiments/problem settings, with all main results and interpretation of results available in the main paper; c) providing ablation experiments, in particular with regards to the regularisation strategy ; c) moving experiments whose results do not fit in the main paper to the appendix, where all details and results can be discussed.

---

> ### Author Response · Authors · 2020-11-17
> **Reply to Reviewer 4**
>
> We would like to thank Reviewer 4 for their thorough and thoughtful comments.
>
> Here we would like to address some of the questions raised by the reviewer:
> * **Incremental novelty**. We agree that the novelty of our approach and what we consider to be particular strengths should be better articulated and a dedicated discussion of main contributions would be a good addition to the paper. While it is true that individual components of our approach are not novel, we think that the main contribution of the paper is a particular model design that is general enough to be used in a variety of different tasks like single-task, multitask learning, domain adaptation and more, while providing means of soft weight sharing that is very simple both in terms of the structure and optimization.
> * **Experimental details**. We agree that the experimental section in the current version of the paper lacks some important detail of the experimental setup. We moved those details to the supplementary material due to the page limit, but we will try our best to better organize the paper so that the essential information is included in the main text.
> * **Model regularisation**.
>   * We agree that the proposed regularization strategy is an important component of the paper and that a careful ablation study would greatly improve the paper. The main reason for not providing this detailed study and moving the description of the method into the appendix is that: (a) this regularizer is optional and in practice we used it in only some of the experiments, (b) having a hard time fitting everything into the main text. We will re-evaluate this decision.
>   * The regularizer is meant to be primarily used in the situations when it is truly important that the identified mixture weights are indeed one-hot (in this case, we can just use templates as is and save computation on mixing multiple templates), or when we need to speed up the convergence of mixture weights. In most of our experiments (Tables 1, 2, 5, 6, 7) we did not use any regularization except for a conventional weight decay. Example of an experiment with a “clustering regularizer” can be found for example in Figure 2b.
>   * To address another question, the templates were not regularised to comprise different sets of weights, and in practice we did not encounter the problem of templates being too similar to each other.
> * **Comparisons to closest works**. We definitely agree that our paper lacks sufficient comparisons to SOTA, or just comparable modular approaches. We will attempt to fix this in the future version of our manuscript.
> * **Domain Adaptation**. In our DA experiments, we do exactly as the Reviewer 4 kindly suggested, which is we train all templates on a number of source domains simultaneously. The smaller the number of available templates, the more parameter sharing naturally appears within different tasks/domains.

---

### Official Review · AnonReviewer2 · 2020-10-28
**Neat but oversold**

**Rating:** 4
**Confidence:** 4

**Review:**

Summary: This paper considers “modular multi-task learning” where parameters in each layer/task are generated as a (layer/task-specific) linear (mixture) combination of a common pool of parameters. Exploring this idea, several observations are made: (1) single task isometric model performance on ImageNet can be improved, (2) Multi-task learning is supported and parameter sharing (selecting same mixture components) emerges in early layers, with specialisation emerging in later layers. With multi-domain learning, the opposite effect is achieved with domain-wise specificity arising in earlier layers, and sharing in later layers. (3) Parameter-efficient transfer learning is supported by fine-tuning the task-specific weights for new tasks. (4) Parameter-efficient domain adaptation is supported by optimising the task-specific weights for new domains.

Overall impression: It’s a neat exploration of parameter sharing in isometric networks. However the novelty is over claimed/prior literature missed, and the experiments are not good enough to make up for this limited novelty with empirical insights discovered or SotA performance reached.

Strengths:
+ Overall the idea is reasonable, and the results make sense.
+ By taking the idea of isometric networks seriously, this paper explores a richer space of parameter sharing than some existing papers that use the same idea.
+ By using soft modularisation, the framework is easier to train than other methods that need to use gradient estimators for hard sharing.
+ It’s good to have a framework with flexibility for single-task, multi-task, and cross-task/domain transfer.

Weaknesses:
1. Novelty/Related work. This paper makes a stronger claim about novelty than is justified.
- The proposed method is a special case of tensor-factorisation based parameter sharing that has already been widely studied in several papers. Specifically, where a vanilla baseline uses a parameter set that can be described as a tensor X of size (W*H*C1*C2*L)  (here L = layers for STL or tasks/domains for MTL), then the proposed method instead uses a parameter set (Y,Z) where X~=Y*Z and Y is a (W*H*C1*C2*K) tensor of K “shared modules” and Z is a (K*L) tensor of “task/layer-specific mixture parameters”, and “*” is the corresponding tensor contraction. This kind setting has already been widely analysed in papers such as [A,B] among others.  The minor difference is the softmax constraint on the factor Z, which gives it the specific “mixture” interpretation, but this was already studied in related methods such as [F].  Parametrically, it also seems to be a special case of methods such as [G].
- With regard to observations about changes in the per-layer sharing strength in multi-task/multi-domain learning, similar observations were made in [A] among others.
- With regard to transfer learning by re-training task-specific factors in such a factorisation setup, this was done by lots of papers including [B,C,D].
- With regard to cross-layer parameter sharing specifically, this is somewhat less commonly studied, but was included in [B,G].

2. Evaluation
- Sec 4.2. It’s neat that increasing layers with a fixed number of parameters can improve ImageNet performance. But this is not quite unprecedented, and it’s not compared with other related approaches to achieving the same such as HyperNets or [B,G] etc. So we don’t know how impressed to be.
- Sec 4.3. Fig 2 is neat. But again, it’s not quantitatively or qualitatively compared to the literature on soft-sharing for deep MTL, which is now very large across both the ML and CVPR communities.
- Sec 4.3. Tab 2 is also not compared to any of the numerous alternative methods for transfer learning out there. Neither in the broader TL field where there are many alternatives, nor among highly related methods such as [B,C,E,etc]

3. Insight. Prior papers that have placed an emphasis on modularity usually resort to significantly more complex optimization strategies [G,etc]. But such papers often start by mentioning and discarding variants of the simple "soft" weighting strategy used here before moving on to more complex setups, e.g., for optimising "hard" module selection. It’s good that this paper makes modularity work with a very easy setup, but it would be good to have more insight on what are the key tricks that are required to achieve it with simple soft blending and optimisation. Presumably something “obvious” was missed by prior work that resorted to much more complex setups.


[A] ICLR’17, “Deep Multi-Task Representation Learning: A Tensor Factorisation Approach”
[B] arXiv’19/AAAI’20 “Incremental Multi-domain Learning with Network Latent Tensor Factorization”.
[C] IJCAI’17, "Tensor Based Knowledge Transfer Across Skill Categories for Robot Control”
[D] IEEE Trans ASL’15, ”Cluster Adaptive Training for Deep Neural Network Based Acoustic Model”
[E] NeurIPS’17, “Learning multiple visual domains with residual adapters”
[F] ICLR’18, “Beyond Shared Hierarchies: Deep Multitask Learning through Soft Layer Ordering”
[G] NeurIPS’19, “Modular Universal Reparameterization: Deep Multi-task Learning Across Diverse Domains”

A. Minor:
- Reference Kirsch 2018 is repeated.
- Fig 2 caption calls it “domain adaptation”, but to my understanding this experiment might better be called multi-domain learning, than domain-adaptation per-se.

---

> ### Author Response · Authors · 2020-11-17
> **Reply to Reviewer 2**
>
> We would like to thank Reviewer 2 for their very thoughtful and detailed review. We were not familiar with the references that the reviewer mentioned, but they are extremely relevant and we will make sure to reflect them in our final publication text. Overall, we agree with the criticism including that the current version of the manuscript could do a much better job in providing a more detailed empirical comparison with the prior work. We also agree that most components of the proposed approach are not novel by themselves. However, in our view, the main strength of the described idea is that it can become a simple, but very general and expressive framework for performing a wide variety of tasks that can benefit from modularity: those discussed in the paper like multi-task learning, transfer learning and domain adaptation and many more that we hope to explore or already explored. While we do not necessarily expect that for any of these tasks in particular, the proposed approach will beat SOTA by a large margin, we hope to alter the manuscript to better substantiate the claim that our method can provide a comparable or better performance in most of them.
>
> We also agree that the simplicity of the optimization for the proposed approach appears to be one of the main (if not the main) strengths. While it is difficult to gain a theoretical insight into the properties of this particular model family (not that we should not try to do this), we think there are a few design choices that may be at least partially responsible for making it work: (a) we observed empirically that the softmax nonlinearity bounding individual mixture weights and ensuring their non-negativity plays an important role and lifting these restrictions can quickly destabilize the model (experiments we probably should have included); (b) highly speculatively, nonlinear dependence of the activations on the mixture weights in contrast to linear dependence in some “mixture of experts” approaches may be more beneficial for model expressivity.

---

### Decision · Program_Chairs · 2021-01-07
**Final Decision**

**Decision:**

Reject

**Comment:**

This paper presents an approach for modular multi-task learning. All the reviewers believe the goals are appealing and the idea is reasonable. However, R2 and R4 raise concerns with respect to novelty. There are also strong concerns regarding experiments. The concerns vary from reproducibility to small improvements and right baselines. The rebuttal fails to provide any new experiments or handle the reviewer concerns. All reviewers and AC agree that paper is not yet ready for publication.